# Effects of Dietary Nitrates on Time Trial Performance in Athletes with Different Training Status: Systematic Review

**DOI:** 10.3390/nu12092734

**Published:** 2020-09-08

**Authors:** Tomáš Hlinský, Michal Kumstát, Petr Vajda

**Affiliations:** Faculty of Sports Studies, Masaryk University, Kamenice 5, 625 00 Brno, Czech Republic; kumstat@fsps.muni.cz (M.K.); vajda@fsps.muni.cz (P.V.)

**Keywords:** nitric oxide, dietary supplements, oxygen consumption, muscle fibres, physical activity

## Abstract

Much research has been done in sports nutrition in recent years as the demand for performance-enhancing substances increases. Higher intake of nitrates from the diet can increase the bioavailability of nitric oxide (NO) via the nitrate–nitrite–NO pathway. Nevertheless, the increased availability of NO does not always lead to improved performance in some individuals. This review aims to evaluate the relationship between the athlete’s training status and the change in time trial performance after increased dietary nitrate intake. Articles indexed by Scopus and PubMed published from 2015 to 2019 were reviewed. Thirteen articles met the eligibility criteria: clinical trial studies on healthy participants with different training status (according to VO_2max_), conducting time trial tests after dietary nitrate supplementation. The PRISMA guidelines were followed to process the review. We found a statistically significant relationship between VO_2max_ and ergogenicity in time trial performance using one-way ANOVA (*p* = 0.001) in less-trained athletes (VO_2_ < 55 mL/kg/min). A strong positive correlation was observed in experimental situations using a chronic supplementation protocol but not in acute protocol situations. In the context of our results and recent histological observations of muscle fibres, there might be a fibre-type specific role in nitric oxide production and, therefore, supplement of ergogenicity.

## 1. Introduction

Nitric oxide (NO) plays a crucial role in signalling and physiological regulatory functions of the human body which are crucial to exercise economy and performance (i.e., vasodilatation, mitochondrial respiration, glucose and calcium (Ca^2+^) homeostasis, skeletal muscle contractility and fatigue development). Because the NO molecule is highly unstable, there is a constant need for its regeneration [1,2,3,4].

Interestingly the substrates for NO syntheses (L-arginine and nitrates) come from our daily diet and therefore can be manipulated [5]. Increased availability of NO via diet has been linked to ergogenic effects [6,7]. However, not all athletes can benefit from these nutritional strategies [8,9]. It seems the effectiveness of NO inducing foods and supplements is limited by an athlete’s training status (aerobic fitness) and muscle fibre type ratio [10].

### 1.1. NO Metabolism and Physiological Importance

Nitric oxide synthesis in the human body is carried out via two pathways: the NO synthase (NOS)-dependent pathway and nitrate–nitrite–NO (NO_3_^−^–NO_2_^−^–NO) pathway [11]. Formation of NO via the NOS-dependent pathway is carried out through the utilisation of L-arginine and oxygen (O_2_). Thus, this reaction relies on the delivery of O_2_ [12]. Insufficient O_2_ delivery during high-intensity exercise may cause this pathway to become dysfunctional [13]. Therefore, the O_2_-independent pathway can substitute NO production [14] via the reduction of NO_3_^−^ from the diet (e.g., green leafy vegetables, beetroot, radish) [15]. Nitrates are reduced by oral anaerobic bacteria to NO_2_^−^ [16]. Subsequently, part of the total NO_2_^−^ is reduced to NO in the acidic environment of the stomach where it protects the organism from some pathogens [17]. The rest of the NO_2_^−^ is then transported via the upper gastrointestinal tract into the blood reaching its plasma peak level 2–3 h postprandially [18]. The reduction of NO_2_^−^ to NO substitutes the O_2_-dependent NOS pathway in various tissues under hypoxic or acidic conditions [19]. These are conditions that typically occur in working muscles during vigorous exercise [20].

Increased NO bioavailability via food or supplements may increase exercise performance, as it is the critical factor in three physiological mechanisms related to physical exercise. Firstly, NO increases skeletal muscle O_2_ delivery via vasodilatation [21]. Secondly, it reduces the O_2_ cost of mitochondrial ATP resynthesis via an increased number of ATP molecules (P) formed per O_2_ molecules consumed (O) in the electron transport chain (P/O ratio) [22], possibly via the interaction with five-coordinated cytochrome C oxidase [23,24]. Lastly, it improves the efficiency of muscle contractility via reduced creatine phosphate (PCr) cost of force production [25] and changes in Ca^2+^ metabolism within the muscle cells [26]. Enhancing these physiological mechanisms leads to more efficient energy metabolism, lower O_2_ demands of the working muscles and higher muscle contractility [26] and therefore an increase in exercise economy and performance [1,22,25].

### 1.2. The Role of Dietary Nitrates in Exercise Physiology

Dietary nitrates (DN) have become a trendy topic in sports nutrition as even minor enhancement of human physiology may positively affect high-intensity exercise and, therefore, competition results. For example, there may be a decrease in the time of time-trial (TT) physical activities [27]. It is also important to note that these changes seem to be highly related to the dosage and supplementation protocol where exercise economy can be improved after a single dose of DN [28,29,30,31,32,33], but exercise performance is more likely augmented after chronic use of DN [34,35,36]. Interestingly, it seems ergogenicity is somehow related to the fibre-type ratio in muscles, augmenting the exercise economy and performance more likely via type II muscle fibres (MFs) than type I [37]. 

During high-intensity exercise, oxidative phosphorylation is diminished as the O_2_ supply is inadequate, and reliance on the anaerobic metabolic pathway of ATP regeneration is favoured [38]. Long-duration high-intensity exercise and intermittent high-intensity exercise lead to exercise-induced hypoxemia causing the muscles to become hypoxic and acidic [39]. This impairment in homeostasis may also disrupt the functioning of the NOS-dependent pathway, and continuation of exercise is highly dependent on the activity of type II MFs [40,41]. Therefore, the reliance on the substitutional NO_3_^−^–NO_2_^−^–NO pathway independent of O_2_ supply is increased [40,42]. Moreover, type II MFs have a lower blood supply which affects the partial pressure of O_2_ within the microvasculature (P_mv_O_2_) causing a lower O_2_ supply compared to type I MFs [43,44,45]. This phenomenon underlines the reliance on the NO_3_^−^–NO_2_^−^–NO pathway in type II MFs and even more under hypoxia [24,46]. Lastly, most recent studies suggest improvement in muscle force production and mitochondrial oxidative phosphorylation are more likely observed in type II MFs than in type I MFs after DN supplementation [47,48].

Furthermore, it has already been suggested in earlier studies that neither acute nor chronic DN supplementation can improve performance in highly trained cyclists [49,50,51], runners [52,53] or cross-country skiers [54]. These groups of endurance-trained athletes tend to have a higher type I MF ratio [55,56,57]. In contrast, high doses of DN (8.4–9.6 mmol) improved performance in highly trained kayakers and rowers [29,58] but not low doses (~4–5 mmol) [29,59]. In this context, the upper body muscles (e.g., biceps brachii, triceps brachii, deltoid, trapezius or latissimus dorsi) are well described as muscles with a higher type II MF ratio [60,61,62]. Moreover, as highly trained athletes develop specific adaptation to rowing [63] and kayaking [64], increased type II MF ratio or MF hypertrophy is more likely [65]. This exercise modality, which mostly involves muscle groups with a higher type II MF ratio [65], can be another example of the fibre-type specific effects of DN [37,66].

### 1.3. Training Status as a Limiting Factor

Nutritional strategies to increase the bioavailability of NO and possibly physical performance have been under the scope of research for many years, and there are some crucial variables (e.g., muscle fibre-type ratio, physiological limitations) which can influence their effectiveness [8,67]. Consumption of dietary NO_3_^−^ (DN) in the form of either nitrate-rich foods or supplements generally increases plasma levels of NO_2_^−^ [28] which interestingly does not always lead to improvement in exercise performance especially in well-trained endurance and elite endurance athletes [68].

Fibre-type specific effects of DN supplementation have been demonstrated in animal experiments where muscle force development increased in type II (fast-twitch glycolytic fibres type IIx) but not in type I MFs (slow-twitch oxidative fibres) [34]. Reliance on the NO_3_^−^–NO_2_^−^–NO pathway is higher in type II MFs due to the lower O_2_ tension (pO_2_) than in type I [43]. Therefore, an increase in the bioavailability of NO mainly affects type II MFs [37]. Endurance-trained athletes are likely to have a higher ratio of slow oxidative type I MFs compared to non-trained and recreationally active athletes [69] or the inactive population and the elderly [70]. This phenomenon also relates to higher aerobic fitness in highly and elite trained athletes which cannot be augmented any further [68]. These seem to be possible explanations for the lower ergogenicity of DN supplementation in highly trained or elite athletes, as some studies failed to enhance the performance of the participants [52,71]. It has been suggested that the efficiency of DN supplementation is related to an athlete’s training status, especially in high-intensity endurance exercises, e.g., time trial (TT) performance, where O_2_ delivery is impaired and reliance on type II MFs is higher [68].

### 1.4. Aim and Purpose of the Review

This article reviews the relationship between VO_2max_ and DN ergogenicity in TT performance, defined as the athlete’s training status. We further discuss the relationship between simulated high altitude and DN ergogenicity in the context of increased reliance on type II MFs under hypoxic conditions, as it has been proposed that the DN ergogenicity is fibre-type related. Additionally, we wanted to stress the importance of inclusion of certain methodological tools in the evaluation of the results of experiments (e.g., smallest worthwhile change and typical error of measurement).

## 2. Materials and Methods 

A review of the literature was carried out using articles indexed by Scopus and PubMed published from 1 January 2015 to 31 December 2019. The PRISMA guidelines were followed to process the review [72].

### 2.1. Search Strategy, Data Extraction and Analysis

The following terms were used for the search: dietary nitrates, physical performance and physical exercise. All descriptors were searched using the Boolean operators to maximise the search quality as follows: *((dietary AND nitrates) AND ((physical AND exercise) OR (physical AND performance)))*. The title and abstract of each article were assessed for eligibility. After that, full-text articles were reviewed to determine their suitability for inclusion or exclusion. The inclusion criteria were: (i) studies on healthy human subjects, (ii) original, English-language research articles, (iii) inclusion of the training status of participants (VO_2max_, training load or other affiliation to training status group), (iv) experimental protocol consisting of time-trial testing, (v) results showing a change in time trial performance, (iv) supplementation protocol is described. The exclusion criteria were: (i) the above inclusion criteria were not met and (ii) the experimental test was combined with environmental modification (e.g., high-altitude simulation and high temperature). No restrictions were made on age, sample size, supplementation protocol or supplement form. Figure 1 describes the process of selecting the articles presented in this review using the PRISMA Flow Diagram.

### 2.2. Data Classification

Out of the 13 included studies, we analysed 22 separate DN experimental situations. The experimental situations were classified into five groups according to the mean group VO_2max_ or VO_2peak_ of the participants according to the preliminary testing of each study (Table 1), which allowed us to define their training status as previously published [73,74]. One study did not use the VO_2max_ of participants for training status assessment but used training load instead [75]. Mean percentual change in time-to-completion of given exercise tasks in the experimental DN groups were either included in the article or calculated from the results.

### 2.3. Quality Assessment

Risk of bias was assessed using the Physiotherapy Evidence Database (PEDro) scale, which provides a reliable assessment of internal validity [76,77]. Each article was individually assessed by the reviewer (T.H.) using an adjusted 11 item checklist to yield a maximum score of 10.

### 2.4. Data Extraction

Data were extracted and reviewed by one of the researchers (T.H.). Additionally, the data were independently reviewed by the other researchers (M.K. and P.V.), following a systematic selection list which included the inclusion criteria mentioned in Section 2.1.

### 2.5. Statistical Analysis

A one-way ANOVA was conducted to determine whether there is a relationship between DN ergogenicity and training status level. Data were classified into three groups according to the training status of participants: untrained + recreationally trained (*n* = 5), *trained* (*n* = 4), *well-trained + elite* (*n* = 13). There were no outliers as assessed by a boxplot. Data were normally distributed for each group as assessed by the Shapiro–Wilk test (*p* > 0.05), and variances were homogeneous as assessed by Levene’s test of homogeneity of variances (*α* = 0.05). A Spearman’s rank-order correlation and Pearson correlation coefficient were used to assess the relationship between DN ergogenicity and VO_2max_ after chronic/acute DN intake and between DN ergogenicity and TT test duration. A coefficient of variation (CV) was carried out to assess the frequency distribution of VO_2max_ among the groups of participants and inter-study variability in performance effect. All data analyses were performed using the SPSS statistical package (version 25; SPSS, Chicago, IL, USA) and the GraphPad Prism 8 (GraphPad Software, San Diego, CA, USA). Results are presented as mean ± standard deviation.

## 3. Results

The reviewed articles included 22 experimental situations presented in Figure 2. The figure shows performance changes in all acute (blue bars) and chronic (red bars) experimental situations reviewed. The effect of DN supplementation is displayed as the percentual mean change in the experimental group. Improvement in performance is interpreted as a negative value as the TT test was finished faster compared to placebo.

The mean changes in performance across the selected experimental situations were −2.63%, −2.83%, −0.49%, −0.27% and −0.31% in untrained (*n* = 1), recreationally trained (I = 4), trained (*n* = 4), well-trained (*n* = 12) and elite (*n* = 1) groups, respectively.

One-way ANOVA presented a statistically significant difference in performance change among groups with different training status after DN supplementation *F* (2, 19) = 11.787, *p* = 0.001 and ω^2^ = 0.533. The positive performance change increased from well-trained + elite (−0.272 ± 0.671%) to trained (−0.490 ± 1.238%) and untrained + recreationally trained (−2.786 ± 1.629%) groups, in that order. Tukey’s post-hoc analysis revealed that the increase from well-trained + elite to the untrained + recreationally trained group (2.51, 95% CI (1.12 to 3.91)) was statistically significant (*p* = 0.001), as well as the increase from trained to untrained + recreationally trained (2.30, 95% CI (0.52 to 4.07), *p* = 0.010), but not from well-trained + elite to trained. Statistical analysis was done on a sample of all 22 experimental situations. Therefore, the data were divided according to the training status but not according to the supplementation protocol due to the small sample size for ANOVA analysis.

A Spearman’s rank-order correlation was run to assess the relationship between VO_2max_ and DN ergogenicity in the TT tests after chronic intake of DN. There was a statistically significant, strong positive correlation between VO_2max_ and performance change in TT, i.e., a positive ergogenic effect declines with higher training status (*r_s_*(9) = 0.810, *p* = 0.003; Figure 3.). However, Pearson’s correlation coefficient showed no significant correlation for acute supplementation (*r*(8) = 0.445, *p* = 0.198; Figure 4). One study did not report VO_2max_ values of participants and, therefore, was excluded from the analysis [75].

### Methodological Quality of Studies

All the reviewed studies that met the eligibility criteria scored 10 out of 10 in PEDro scale as they followed double-blinded crossover placebo-controlled experimental design. Overall, the quality of the reviewed studies was assessed as excellent.

## 4. Discussion

We reviewed the available research from 2015 to 2019 focused on the relationship between dietary nitrates and TT performance in athletes with different training status according to their VO_2max_. Statistical analysis of the data collected from 13 articles showed that the higher the training status, the lower the effect on TT performance.

### 4.1. Dietary Nitrates and Time Trial Performance

Our results demonstrate that DN ergogenicity in TT performance is more likely observed in less-trained individuals than in the highly trained and more consistently after chronic use. These results are in agreement with previously reported data [1]. In contrast, some reviews did not find a statistically significant change in TT performance [86,87]. It was proposed that the VO_2max_ > 65 mL/kg/min is not related to ergogenic effects after DN intake [1,10]. However, our results suggest that DN ergogenicity may be lower even in trained athletes, as we did not find a significant difference between the trained and well-trained + elite groups. Significant ergogenic effects were evident in the groups of VO_2max_ ≤ 50 mL/kg/min and less.

These results suggest there is a strong link between the athlete’s training status and DN ergogenicity. The real cause of this phenomenon is still debated. We further provide a discussion on possible causes.

### 4.2. Dietary Nitrates and Exercise Performance in a High-Altitude Environment

Our results are consistent with other studies suggesting the ergogenicity of DN is diminished in highly-trained endurance athletes, possibly due to the fact of their higher type I MF ratio [1,37,88].

To support the hypothesis of a relationship between ergogenicity and type II MFs, we also reviewed TT studies at simulated high-altitude through normobaric hypoxia (*n* = 3; Figure 5). A high altitude environment with lower pO_2_ leads to impairment in O_2_ delivery in tissues, and the aerobic energy metabolism pathway is diminished [89]. Therefore, the work efficiency of type I MFs is lowered, and reliance on the anaerobic metabolic pathway and type II MFs is increased [90]. Exercise-induced hypoxemia is amplified in high-altitude hypoxic conditions, and the reliance on type II MFs is expected to be increased [91,92]. Therefore, it has been hypothesised that DN ergogenicity in highly-trained endurance athletes could be augmented due to the increased involvement of type II MFs during TT tests in an environment with lower p0_2_ [93,94,95]. Although performance improvement was observed in highly-trained athletes (VO_2max_ > 65 mL/kg/min) after DN intake (−3.2%) [94], yet again, more consistent improvement was more likely observed in less-trained athletes (−3.6 ± 0.4%) [93,94].

Additionally, Arnold et al. [95] observed an improvement in the highly trained (−0.4%), but this was not of statistical significance (*p* = 0.6). To this day, data on the ergogenicity of DN at high altitude brings more or less mixed results and, so far, no additional benefits in general [96,97]. Despite the increased pulmonary NO availability and plasma NO_2_^−^ concentrations [98] and lower altitude-induced reduction in endothelial function [99] that have been documented, solid conclusions as to whether a high altitude could augment ergogenicity in highly-trained athletes requires further research.

### 4.3. Dietary Nitrates Studies Methodology

Nyakayiru et al. [81] observed a statistically significant and positive response in six out of the total of 17 participants during TT performance. Whereas the mean VO_2max_ (65.0 ± 4 mL/kg/min) classified the group as well-trained, the standard deviation suggested that some of the participants were less trained and potentially more likely to be DN responders. In future research, we suggest the grouping of participants to be carefully divided according to training status, because there seems to be a thin line between the high and low ergogenicity of DN and, therefore, statistical evaluation of the results may be affected.

Correctly classifying training status (e.g., VO_2max_ and/or weekly training load) in various athletes may be an essential aspect of increasing the reliability of DN research [73,74]. Table 2 summarises the research characteristics of the reviewed studies, showing that specific TT criteria (distance, test duration) and DN supplementation protocols (acute/chronic, DN dosage) are different across the studies. Notably, groups of participants are somewhat mixed in terms of training status as the diversity of the athlete’s VO_2max_ is in some studies quite high (CV = 21.5% in Oskarsson and McGawley [82]).

Twenty-two experimental situations were included in our review with highly variable TT test durations 881 ± 856 s (25–3300 s). No correlation was observed between the TT duration and performance effects (*r_s_*(20) = −0.237, *p* = 0.288). There was also no correlation after the TT duration adjustment (i.e., only situations with TT duration > 10 min included; *r_s_*(10) = 0.064, *p* = 0.843). However, it may be speculated whether long-duration TT tests (>720 s) could benefit more from DN supplementation as the authors of 10 out of the 12 experimental situations reported ergogenic effects. In contrast, only 5 out of 10 experimental situations (TT test duration < 720 s) reported improvement in exercise performance. It is becoming more evident that DN ergogenicity could be fibre-type specific. Therefore, the duration of the event should be taken into consideration as our results are in contrast with general recommendations for DN use (4–8 min events in Burke et al. [2]).

Inter-study variability in performance effect was high as the coefficient of variation (CV) reached 55%. The lowest and therefore more stable CV was obtained in trained (26%) and well-trained groups (26%) but not in the recreationally trained groups (58%).

Dosage, time-to-test intake, supplementation protocol and form of DN also differed across the reviewed studies. In eight studies, the dosage exceeded the amount of 8 mmol of NO_3_^−^ (up to 12.8 mmol; *n* = 5) which is the generally recommended amount whether an acute or chronic protocol is chosen [6]. Time-to-test intake is 2–3 h in most studies (*n* = 11). In contrast, in some studies participants were instructed to consume the supplement less close to the test (3.5 h in Porcelli et al. [68]) or closer to the test (1 h in Callahan et al. [79]). A more consistent positive change in performance is likely observed in athletes following a chronic supplementation protocol [1]. Our results showed a strong positive correlation (*r_s_*(9) = 0.810, *p* = 0.003) between the VO_2max_ and performance change after chronic DN intake but no correlation in acute intake studies (*r*(8) = 0.445, *p* = 0.198). This suggests that chronic DN intake provides more consistent results for all training status categories and underlines the lower ergogenicity in highly trained athletes. The most common form of DN supplementation is beetroot juice concentrate (BJC) [27,58,71,78,82,83,84]. Other forms of DN supplementation include raw beetroot juice (BJ) [85], encapsulated potassium nitrate (PN) [75,80] and beetroot crystals (BC) [79] or a water solution containing sodium nitrate (SN) [68,81]. All these variables in the supplementation protocol could potentially interfere with the ergogenic effects. It has been well documented that individual pharmacokinetic responsiveness after DN intake varies [10,28,31].

### 4.4. The Real Benefit for Elite Athletes?

Significant beneficial effects of DN intake in elite endurance athletes are less likely to be observed. This athletic group demonstrated a high proportion of type I MFs, elite exercise performance close to the athlete’s physiological limits or NO_3_^−^-mediated vasodilation in non-prioritized muscles which may lead to reduced O_2_ delivery to the essential muscles working very close to their maximal cardiac output [100]. All these factors are now suggested as potential causes of the lower ergogenicity in highly trained athletes.

Although the available laboratory studies do not support DN ergogenicity in well-trained athletes by providing statistically significant outcomes, some might argue whether this is of concern in real sport events. From the perspective of more ecologically valid situations (e.g., sports competitions), even statistically insignificant improvement can change the overall result of TT performance. Therefore, it may be speculated whether the 0.63% change in performance found in our analysis in the well-trained group [27] could change the outcome in a real sports event. Other studies have discussed the same phenomenon. For example, Rokkedal-Lausch et al. [27] concluded: “The improvement in 10-km TT completion time and power output of 0.6% and 1.6%, respectively, in the present study, is of practical relevance for elite and well-trained athletes”. Seemingly, the improvement in performance of only 0.6% could make a significant difference when comparing the TT durations of elite cyclists. For example, during the 17 km TT at the 2019 Giro d’Italia (Verona-Verona, winning time 22:07) or during the 31 km TT at the 2019 Tour De France (Saint pée sur Nivelle–Espelette, winning time 40:52) only 0.3% and 0.04%, respectively, separated first and second position [101,102]. Nevertheless, in the laboratory environment, the smallest worthwhile change (SWC) and the typical error of measurement (TEM) should be taken into consideration when interpreting the results [103,104]. According to Hopkins [103], the improvement in an athlete’s performance cannot be considered as progressive until it reaches or goes over 0.3% in individual sports, defining this improvement as the SWC. Additionally, the TEMs for TT cycling tests are usually around ~1% depending on the ergometer and time variation for short (~60 s) and long-duration TT (~1 h) in elite athletes, also around ~0.5% and ~1%, respectively, depending on the method and the equipment [105]. Furthermore, if we would focus on a longer duration TT test (several minutes) then the athlete’s improvement would have to reach 1.3% (SWC + TEM), so it could be called beneficial [103]. Therefore, even the most considerable TT performance changes observed during cycling ergometry measurements in elite athletes may not be relevant (improvement is lower than SWC + TEM, therefore unclear) in real sports events [103]. Interestingly only 5 of the 13 reviewed studies observed performance improvement after DN supplementation ≥ 1.3% [68,75,81,84,85].

From this point of view, it is evident that translating athlete-specific DN research outcomes into practical interventions requires the determination of their true translational potential. To standardise this, Close et al. [106] recently proposed an excellent 9 step framework, that may assist practitioners in the proper evaluation of performance nutrition research and applying the findings into practice therefore making the dietary choices adopted by their athletes sport-specific and “evidence-based”.

### 4.5. Future Perspectives

Recent results from muscle histology studies show fascinating new insights into the role of NO_3_^−^ in the human body [107,108,109,110,111]. Seemingly NO_3_^−^ can be stored within the muscle tissue cells which could be a very elegant explanation of the crucial role of muscle tissue in NOS production, the NOS-dependent and NO_3_^−^–NO_2_^−^–NO pathways [88]. Wylie et al. [66] suggested that there is a relationship between VO_2max_ and NO_3_^−^ substrates within the MFs. In the light of these recent findings, significant improvements in TT performance in athletes with VO_2max_ < 65 mL/kg/min may be explained by lower muscle NO_3_^−^ disposal and NO bioavailability. Therefore, increased DN intake may increase muscle NO_3_^−^ storage capacity and positively affect physical performance [112]. Moreover, it has been hypothesised there could be a potentially positive effect in the treatment of cardiovascular and metabolic diseases as NO_3_^−^ levels are lower in the elderly, untrained and cardiovascular or diabetes patients [111,112,113]. Therefore, further research in the area of NO_3_^−^ muscle storage and its role not only in physical exercise but also in therapeutic practice is needed.

Based on the above discussion, it remains unclear whether any statistically insignificant changes in highly trained athletes could be affected by MF ratio, non-specific tissue vasodilatation, physiological systems already at their maximal limit of adaption in elite athletes, supplementation protocol, NO_3_^−^ muscle storage or other factors. As such, future research should focus on unifying the study design framework and strict classification of participants according to their training status. Additionally, in the light of recent observations of the fibre-type specific effects of DN, it would be reasonable to focus more on resistance and speed exercise tasks or lactate-anaerobic tasks to compare DN effectiveness among different sports disciplines.

### 4.6. Limitations of the Study

Our study has limitations which should be noted. Firstly, we only focused on articles published from 2015 and 2019, as we wanted to review the most recent studies and put them in the context of future perspectives as the NO_3_^–^ muscle storage presents a new and fascinating approach in explaining the DN ergogenicity. Secondly, our study did not focus on data related to NO_3_^−^/NO_2_^−^ blood levels which are the critical substrates for NO production and bioavailability. We managed to associate performance enhancement with chronic use of DN. However, we may speculate whether increased levels of NO_3_^−^/NO_2_^−^ cause such a performance effect. We had not made the statistical analysis of biochemical data to verify such association (some studies do not provide such data as the authors did not focus on the biochemical analysis). Lastly, we analysed mean percentual changes in the performance of the experimental groups gathered from the selected studies resulting in a certain bias as the results can deviate across the group of participants.

## 5. Conclusions

In conclusion, reviewed studies from 2015 and 2019 show that ergogenicity of dietary nitrates in time trial performance is more likely to be observed in lesser-trained athletes. Our results suggest that the higher the athlete’s training status, the lower the exercise performance improvement. These results are more consistent in chronic dietary nitrates supplementation studies rather than in studies following an acute supplementation protocol. Furthermore, even a minor and statistically insignificant improvement in performance of around 0.31% could make a difference for the elite trained, as the performance differences between the podium athletes are tight. However, research results in elite athletes are inconsistent (e.g., improvement/impairment, statistically significant/insignificant), and research samples are usually small. The smallest worthwhile change and typical error of measurement should be used for critical assessment when evaluating time-trial research results. The performance change should be considered as beneficial only when compared to the sum of the smallest worthwhile change and typical error of measurement for selected training status groups and laboratory tests.

## Figures and Tables

**Figure 1 nutrients-12-02734-f001:**
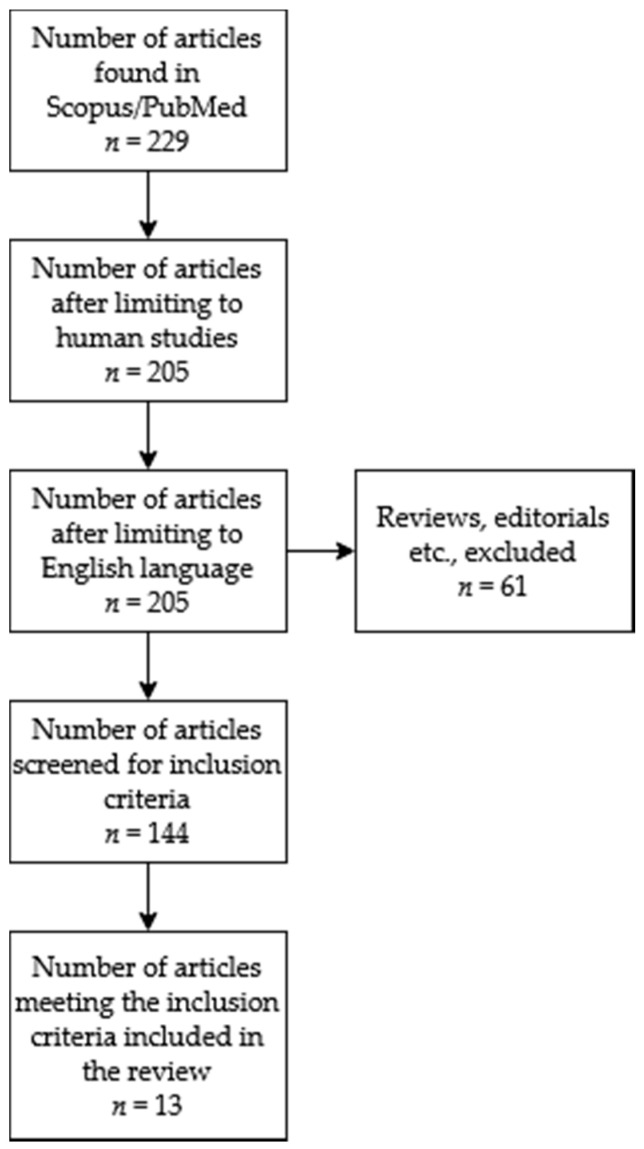
Selection process diagram of articles meeting the eligibility criteria for our review.

**Figure 2 nutrients-12-02734-f002:**
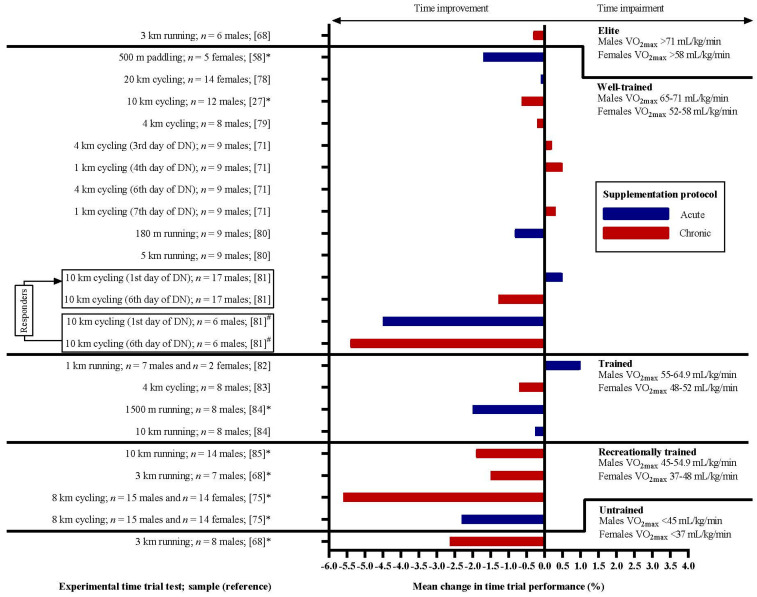
Effects of DN supplementation on TT performance in athletes with different training statuses according to their VO_2max_. The graph presents the mean percentual changes in time-to-completion of given tasks in the experimental groups. Negative values represent improvement, as the task was finished in less time (bars to the left from *y*-axis) following acute (blue bars) or chronic (red bars) supplementation. Cited studies are listed in References [27,58,68,71,75,78,79,80,81,82,83,84,85]. * Statistically significant improvement in performance following DN supplementation; ^#^ improvement was observed in 6 of 17 participants dividing the group of *well-trained* athletes into “responders” and “non-responders”. The two experimental situations are, therefore, divided into four according to the specific reaction to DN supplementation of the participants [81].

**Figure 3 nutrients-12-02734-f003:**
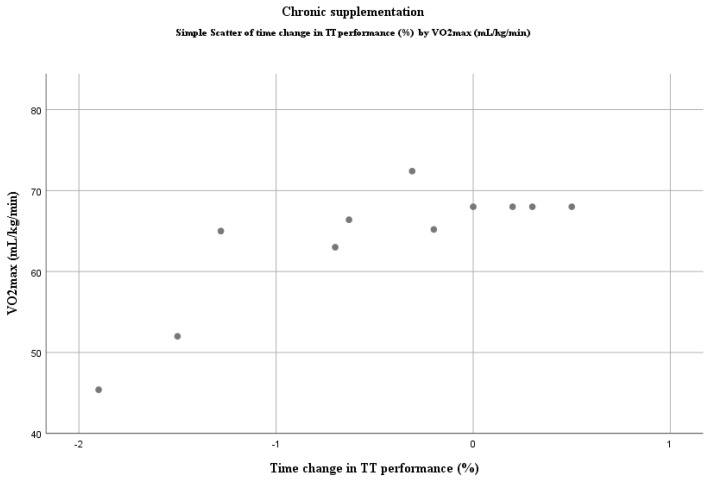
Relationship between VO_2max_ and DN ergogenicity in TT tests: chronic supplementation protocol. Negative values in time change represent improvement as the task was finished in less time.

**Figure 4 nutrients-12-02734-f004:**
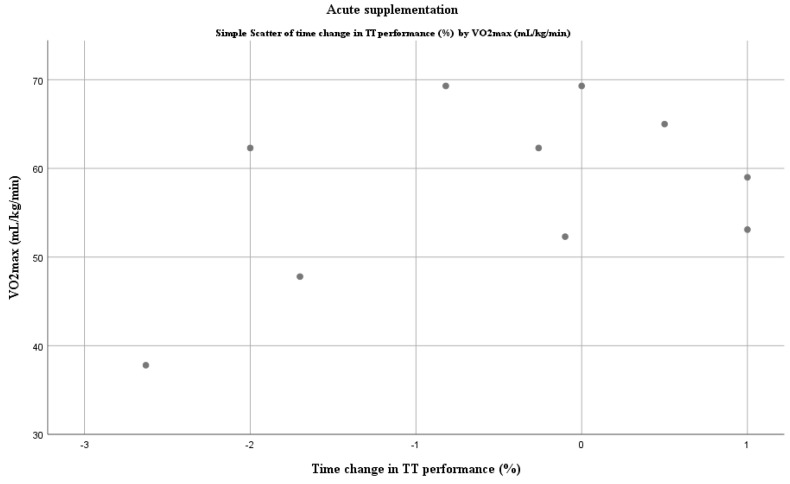
Relationship between VO_2max_ and DN ergogenicity in TT tests: acute supplementation protocol. Negative values in time change represent improvement as the task was finished in less time.

**Figure 5 nutrients-12-02734-f005:**
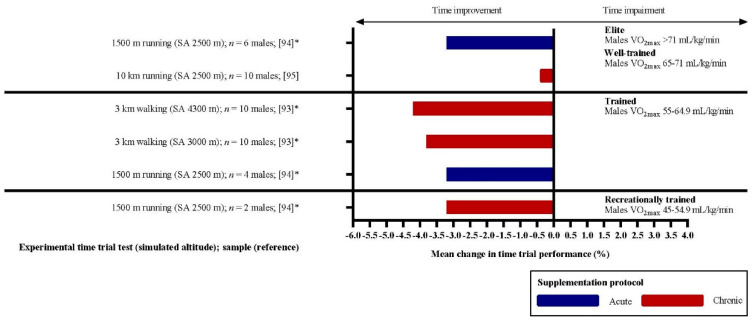
Training status and ergogenicity in simulated high-altitude. Cited studies are listed in References [93,94,95]. * Statistically significant improvement in performance following DN supplementation.

**Table 1 nutrients-12-02734-t001:** Participant training status classification defined by VO_2max_.

Training Status (Number of Experimental Situations Included)	Sex	VO_2max_ (mL/kg/min)
1—Untrained (*n* = 1)	Males	<45
	Females	<37
2—Recreationally trained (*n* = 4)	Males	45–54.9
	Females	37–48
3—Trained (*n* = 4)	Males	55–64.9
	Females	48–52
4—Well-trained (*n* = 12)	Males	65–71
	Females	52–58
5—Elite (*n* = 1)	Males	>71
	Females	>58

**Table 2 nutrients-12-02734-t002:** Relationship between DN supplementation protocol and training status in promoting the ergogenic effects.

Reference	Sex	Training Status	VO_2max_ ^a^/VO_2peak_ ^b^	CV	A/C	Supplementation Protocol	Test (Test Duration)	Ergogenic *
[75]	M and F	Recreationally trained	-	-	Acute	8.0 mmol (NO_3_^−^) of PN; 2 h before the test	8 km cycling (~20 min)	Yes
[84]	M	Trained	62.3 ± 8.1 mL/kg/min ^a^	13.0%		12.8 mmol (NO_3_^−^) of BJC; 3 h before the test	1500 m running (~6 min)	Yes
							10 km running (~45 min)	No
[82]	M	Trained	59.0 ± 2.9 mL/kg/min ^a^	4.9%		6.4 mmol (NO_3_^−^) of BJC; 2.5 h before the test	1 km running (<3 min)	No
	F		53.1 ± 11.4 mL/kg/min ^a^	21.5%				
[80]	M	Well-trained	69.3 ± 5.8 mL/kg/min ^a^	8.4%		9.9 mmol (NO_3_^−^) of PN; 2.5 h before the test	180 m running (~25 s)	No
							5 km running (~17 min)	No
[81]	M	Well-trained	65.0 ± 4.0 mL/kg/min ^b^	6.2%		12.8 mmol (NO_3_^−^) of SN; 3 h before the test	10 km cycling (~17 min)	No ^#^
[78]	F	Well-trained	52.3 ± 4.9 mL/kg/min ^a^	9.4%		6.4 mmol (NO_3_^−^) of BJC; 2.5 h before the test	20 km cycling (~35 min)	No
[58]	F	Well-trained ^†^	47.8 ± 3.7 mL/kg/min ^b^	7.7%		12.8 mmol NO_3_^−^) of BJC; 2 h before the test	500 m paddling (~2 min)	Yes
[68]	M	Untrained	37.8 ± 5.8 mL/kg/min ^b^	15.3%	Chronic	6-day 5.5 mmol (NO_3_^−^) of SN; 3.5 h before the test	3 km running (~15 min)	Yes
[75]	M and F	Recreationally trained	-	-		15-day 8.0 mmol (NO_3_^−^) of PN; 2 h before the test	8 km cycling (~20 min)	Yes
[68]	M	Recreationally trained	52.0 ± 4.5 mL/kg/min ^b^	8.7%		6-day 5.5 mmol (NO_3_^−^) of SN; 3.5 h before the test	3 km running (~12 min)	Yes
[85]	M	Recreationally trained	45.4 ± 5.9 mL/kg/min ^a^	13.0%		3-day 8.4 mmol (NO_3_^−^) of BJ; 2 h before the test	10 km running (~55 min)	Yes
[83]	M	Trained	63.0 ± 4.0 mL/kg/min ^b^	6.3%		8-day 6.4 mmol (NO_3_^−^) of BJC; 2.5 h before the test	4 km cycling (~6 min)	No
[71]	M	Well-trained	68.0 ± 3.0 mL/kg/min ^b^	4.4%		7-day 12.8 mmol (NO_3_^−^) of BJC; 2.5 h before the test	1 km cycling (~80 s)	No
							4 km cycling (<6 min)	No
[79]	M	Well-trained	65.2 ± 4.2 mL/kg/min ^a^	6.4%		3-day ~5 mmol (NO_3_^−^) of BC; 1 h before the test	4 km cycling (<6 min)	No
[81]	M	Well-trained	65.0 ± 4.0 mL/kg/min ^b^	6.2%		6-day 12.8 mmol (NO_3_^−^) of SN; 3 h before the test	10 km cycling (~17 min)	No ^#^
[27]	M	Well-trained	66.4 ± 5.3 mL/kg/min ^a^	8.0%		7-day 12.8 mmol (NO_3_^−^) of BJC; 2.75 h before the test	10 km cycling (~15 min)	Yes
[68]	M	Elite	72.4 ± 6.1 mL/kg/min ^b^	8.4%		6-day 5.5 mmol (NO_3_^−^) of SN; 3.5 h before the test	3 km running (~10 min)	No

^a^ VO_2max_ values; ^b^ VO_2peak_ values; * Statistically significant change in performance; ^#^ ergogenic effects in 6 of 17 participants following supplementation [81]; ^†^ 5 female athletes with VO_2peak_ 47.8 ± 3.7 mL/kg/min (recreationally trained or trained) are presented in the “well-trained” section of this figure because they were described as international-level female kayak athletes and all were 2012 National Squad members with 3/5 athletes competing at the 2012 London Olympic Games [58]. VO_2peak_ could have been lower due to the exercise modality [1]. VO_2max_/VO_2peak_ values are presented as the mean ± SD. Abbreviations: dietary nitrates (DN); male (M); female (F); CV-coefficient of variation; beetroot juice concentrate (BJC); beetroot juice (BJ); potassium nitrate (PN); beetroot crystals (BC); sodium nitrate (SN).

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
