# Peer review of "Effects of Dietary Nitrates on Time Trial Performance in Athletes with Different Training Status: Systematic Review"

_nutrients, 2020, doi:10.3390/nu12092734_

Round 1

Reviewer 1 Report

The author revised manuscript as suggested by Reviewer's comments.

Necessary action has been taken by authors to add referance and text.

Reviewer 2 Report

For the author:

This study aims to evaluate the relationship between the athlete’s training status and the change in time-trial performance after increased dietary nitrate intake. The authors have obviously spent considerable time revising the manuscript and their hard work is clearly paying off. This manuscript is drastically improved from the original submission. The message is very clear, the language is much more clean, and several sections of manuscript (introduction and discussion) are clearly defined. In addition, the methodology carried out has gained in quality thanks to the incorporation the risk of bias analysis. For this reason, I encourage to editor to consider this manuscript for publication for the interesting value of the study realized, that now it is a much more robust study.

This manuscript is a resubmission of an earlier submission. The following is a list of the peer review reports and author responses from that submission.

Round 1

Reviewer 1 Report

For the author:

This study aims to evaluate the relationship between the athlete’s training status and the change in time-trial performance after increased dietary nitrate intake. However, the authors have not clearly showed the existing problems and why it is necessary to carry out this systematic review. This issue must be added in the main document. I have another general question, are others systematic revisions there similar to your study?

For example, after a quick search, I have found the following systematic reviews:

  • The Effect of Dietary Nitrate Supplementation on Endurance Exercise Performance in Healthy Adults: A Systematic Review and Meta-Analysis. (DOI: 10.1007/s40279-016-0617-7) (2017).
  • Nitrate Supplementation Improves Physical Performance Specifically in Non-Athletes During Prolonged Open-Ended Tests: A Systematic Review and Meta-Analysis. (DOI: 10.1017/S0007114518000132) (2018).
  • Effect of Dietary Nitrate Supplementation on Metabolic Rate During Rest and Exercise in Human: A Systematic Review and a Meta-Analysis. (DOI: 10.1016/j.niox.2016.01.001) (2016).
  • The Effect of Nitrate Supplementation on Exercise Tolerance and Performance: A Systematic Review and Meta-Analysis. (10.1519/JSC.0000000000002046). (2018)

What does scientific evidence provide your systematic review that these reviews do not offer?

General Comments:

Introduction: The authors have not generated a relationship between Nitric oxide and its performance effect. You must add more information on this topic and explain what type of physiological pathways are interacting (during exercise).

Methods: I am pleased to know that your systematic search has been carried out in the “Pubmed” database, since it is one of the most important databases. I would like to know the true reason for the search time (from 1. 1. 2015 to 31. 12. 2019) that the authors have used in their review. The figure 1 is difficult to understand; Authors must use the PRISMA traditional model (from top to bottom). Why was not the revision registered in the international prospective register of systematic reviews (Prospero)? In addition, the authors did not carry out the Assessment of Risk of Bias, why not? This issue must be to add in the main document.

Results: The beginning of the results should generate a more extensive explanation, where is it treated in higher depth the diagram for the search strategy? 

Discussion: I do not understand the meaning of this section "The Role of Dietary Nitrates in Exercise Physiology", since the authors do not generate a debate between their results and the data of other authors. Authors simply present the results of other authors. However, the discussion section has another meaning. On the other hand, the discussion section must be structured as regards the training status of participants, for instance, “the benefits of nitrate consumption in trained subjects”, “the benefits of nitrate consumption in well-trained subjects”…. and the obtained results must be discussed with the data already existing by other authors.

What were the Strengths and limitations of this systematic review?

Minor comments

P0 L9-23: VO2max, it could be VO2max .

P0 L27-31: Add more add bibliographic citation.

Reviewer 2 Report

Page no. 492-494: Check the reference style which is wrongly played as per journal style. No need to put all initial letters in capital. Check all references in the manuscript and correct it.

Page 181-183: Improvement in these……………………exercise economy and exercise performance. It shroud be “Improvement in these……………………exercise economy and performance.”

Page 346-348: Need to correlated with bioavailability of nitrates and their performance with different training status.That would be added more advantage to reader.

Round 2

Reviewer 1 Report

RESPONSE TO REVIEWERS COMMENTS

Effects of dietary nitrates on time-trial performance in athletes with different training status: Systematic review

We want to thank the reviewers very much for comments and suggestions. We followed and applied the issues suggested by the reviewer to enhance the clarity. In coordination with all co-authors, we made additional amendments with no changes in the content. All the amendments are highlighted in the revised manuscript.

Reviewer 1

Point 1:

For the author:

This study aims to evaluate the relationship between the athlete's training status and the change in time-trial performance after increased dietary nitrate intake. However, the authors have not clearly showed the existing problems and why it is necessary to carry out this systematic review. This issue must be added in the main document.

Response 1: We added a section 1.4 Aim and Purpose of the review. Text is as follows: " This article reviews the relationship between training status defined by the athlete’s VO2max and DN ergogenicity in TT performance. We further discuss the relationship between simulated high altitude and DN ergogenicity in the context of increased reliance on type II MFs under hypoxic conditions as it has been proposed that the DN ergogenicity is fibre-type related. Additionally, we wanted to stress the importance of inclusion of certain methodological tools in the evaluation of the results of experiments (e.g. smallest worthwhile change and typical error of measurement)."

Importance of this review in points:

  • Relationship between training status and DN ergogenicity
  • Support the fact that DN ergogenicity is fibre-type related.
  • Point out on the need of inclusion of SWC and TEM in the evaluation of results as the change may not be relevant in real-life events of some athletes.

The aim and hypothesis of the study should be added in the text of the introduction. It is not necessary to made a individual section for this purpose.

Point 2: I have another general question, are other systematic revisions there similar to your study?

For example, after a quick search, I have found the following systematic reviews:

Study 1. The Effect of Dietary Nitrate Supplementation on Endurance Exercise Performance in Healthy Adults: A Systematic Review and Meta-Analysis. (DOI: 10.1007/s40279-016-0617-7) (2017).

Study 2. Nitrate Supplementation Improves Physical Performance Specifically in Non-Athletes During Prolonged Open-Ended Tests: A Systematic Review and Meta-Analysis. (DOI: 10.1017/S0007114518000132) (2018).

Study 3. Effect of Dietary Nitrate Supplementation on Metabolic Rate During Rest and Exercise in Human: A Systematic Review and a Meta-Analysis. (DOI: 10.1016/j.niox.2016.01.001) (2016).

Study 4. The Effect of Nitrate Supplementation on Exercise Tolerance and Performance: A Systematic Review and Meta-Analysis. (10.1519/JSC.0000000000002046). (2018)

Point 3: What does scientific evidence provide your systematic review that these reviews do not offer?

Response 2 and 3: Our aim was not to provide a meta-analysis, but a systematical review of recent literature focused on high-intensity exercise in the context of training status. Our study is similar to Study 1 in the way of analysis of TT performance results. However, our study brings more light into the context and relationship between these results and VO2max and discuses the possible explanations of this relationship as the higher aerobic capacity is also linked with certain muscle fibre type ratio.

A systematic review presents descriptive evidences without statistical analysis. However, the meta-analyzes, beside to have descriptive evidences, also provide a quantitative review through statistical techniques. Therefore, a meta-analysis is a more complex study than a systematic review. Furthermore, your study groups a number of 13 articles, however, the study number 1 (DOI: 10.1007/s40279-016-0617-7) groups 28 studies, beside 22 time to exhaustion studies and 8 graded-exercises test studies. Thus, this study already published (DOI: 10.1007/s40279-016-0617-7) is more original, complete and of higher quality. Sorry, but I do not consider your systematic review relevant for publication in this journal, mainly because there are already published works with this topic. The authors should have carried out a first search to check if the topic they wanted to work on was already done.

All the studies mentioned above focus on the relationship between DN intake and physical performance. However, we felt a strong urge to stress the fact that future studies should focus more on the methodology of evaluation of results and relationship between muscle fibre ratio, NO3- muscle storage and DN ergogenicity.

General Comments:

Point 4: Introduction: The authors have not generated a relationship between Nitric oxide and its performance effect. You must add more information on this topic and explain what type of physiological pathways are interacting (during exercise).

Response 4: Relationship between DN supplementation and exercise physiology was moved from Discussion to Introduction, so it is clear enough from the beginning of the article.

Point 5: Methods: I am pleased to know that your systematic search has been carried out in the "Pubmed" database, since it is one of the most important databases. a) I would like to know the true reason for the search time (from 1. 1. 2015 to 31. 12. 2019) that the authors have used in their review. b) The figure 1 is difficult to understand; Authors must use the PRISMA traditional model (from top to bottom). c) Why was not the revision registered in the international prospective register of systematic reviews (Prospero)? d) In addition, the authors did not carry out the Assessment of Risk of Bias, why not? This issue must be to add in the main document.

Response 5 a): We wanted to provide a review of the most recent literature in the context of the body of knowledge in nitric oxide and increasing its bioavailability. Since the true reason of lower ergogenicity of DN is still debated (e.g. NO3- muscle storage, muscle fibre ratio) we wanted to contribute to this debate with a focus on time-trial performance measurements as they are usually performed in high intensity with higher involvement of type II muscle fibres. Type II muscle NO3- storage is yet another key explanation of DN ergogenicity, which is debated for the last decade, but the most important discoveries have been made during the past five years (2015). That is one of the reasons why we had chosen this time period.

The search date should be provided based on the last revision that was made on this topic. If the authors detect previous reviews of 5 or more years, it would be interesting to update the data on this topic. However, as I have said in previous paragraphs, there is a meta-analysis dated on 2017 that has more references.

Response 5 b): Mistake was corrected.

Response 5 c): Thank you very much for this comment. To be honest, we were not familiar with Prospero register, and therefore this is a beneficial suggestion for our work in the future.

You have to take into account this detail for future studies.

Unfortunately, as we have just now learned this register is meant for reviews that are at the beginning of the whole process and not for complete reviews. Therefore, we are not sure if we can correct this mistake.

Figure 1 Source: https://www.crd.york.ac.uk/PROSPERO/#aboutregpage

Response 5 d): We informally assessed the risk of bias by including only articles conducting double-blinded crossover placebo-controlled experimental studies, but we did not include this information in the text. Therefore, the Assessment of Risk of Bias was carried out using PEDro assessment guideline, which was added to the manuscript.

It must be treated in greater depth, it needs more information. For future studies, use the RevMan 5.3 application.

Point 6: Results: The beginning of the results should generate a more extensive explanation, where is it treated in higher depth the diagram for the search strategy?

Response 6: Corrected. We provided more extensive explanation of the diagram.

Nevertheless, the explanation is provided in the note of Figure 2., but if the explanation should be preferably in the text for better clarity than we gladly accept this comment.

It must contain a greater number of sections, so the reader wants to replicate this search, it is easy to carry out.

Point 7: Discussion: a) I do not understand the meaning of this section "The Role of Dietary Nitrates in Exercise Physiology", since the authors do not generate a debate between their results and the data of other authors. Authors simply present the results of other authors. However, the discussion section has another meaning. b) On the other hand, the discussion section must be structured as regards the training status of participants, for instance, "the benefits of nitrate consumption in trained subjects", "the benefits of nitrate consumption in well-trained subjects"…. and the obtained results must be discussed with the data already existing by other authors.

Response 7 a): Thank you for this comment. We moved this section into section Introduction, as it is more of a theoretical nature. Furthermore, we created

Response 7 b): Discussion focuses more on the general issues related to low ergogenicity among the highly-trained athletes. Therefore, it is not divided into sections each related to different training status groups more likely divided into sections each focusing on limitations of DN use in real-life situations and then discussed in the context of typical error of measurement and smallest worthwhile change.

Changes in performance are statistically significant in less-trained athletes, and this phenomenon correlates with VO2max. However, the true nature of limitation of DN ergogenicity might be even more complex as the recent studies point out on the relationship between the NO3- muscle storage etc.

Aim of this review was to stress these questions which have not been answered yet and provide opinion on future focus in nitric oxide science.

Point 8: What were the Strengths and limitations of this systematic review?

Response 8: We added new section 4.6 Limitations of the study. Text is as follows: " Our study has limitations which should be noted. Firstly, we only focused on articles published from 2015 and 2019 as we wanted to review the most recent studies and put them in the context of future perspectives as the NO3- muscle storage presents a new and fascinating approach in explaining the DN ergogenicity. Secondly, our study did not focus on data related to NO3-/NO2- blood levels which are the critical substrates for NO production and bioavailability. We managed to associate performance enhancement with chronic use of DN. However, we may speculate whether increased levels of NO3-/NO2 cause such a performance effect. We had not made the statistical analysis of biochemical data to verify such association (some studies do not provide such data as the authors did not focus on the biochemical analysis). Lastly, we analyse mean percentual changes in the performance of the experimental groups gathered from the selected studies resulting in a certain bias as the results can deviate across the group of participants."

Minor comments

Point 9: P0 L9-23: VO2max, it could be VO2max .

Response 9: Mistake was corrected.

Point 10: P0 L27-31: Add more add bibliographic citation.

Response 10: Corrected. We added a reference to work of nobel prize scientists Ignarro, Murad and Furchgott.
